# The Effect of a Short Course on a Group of Italian Primary School Teachers' Rates of Praise and Their Pupils' On-Task Behaviour

Francesco Sulla [1,*] and Dolores Rollo [2]

1   Department of Humanities, University of Foggia, 71122 Foggia, Italy
2   Department of Medicine and Surgery, University of Parma, 43121 Parma, Italy
*   Correspondence: francesco.sulla@unifg.it

**Abstract:** Teachers who continually experience behaviour problems in their classrooms may believe they are ineffective at working with children. Indeed, student misbehaviour is often identified as the main cause of teacher stress. Strategies designed to increase teacher rates of approval in their class have been shown to be effective in decreasing students' inappropriate behaviours. In this investigation, a group of Italian primary school teachers took part in a brief 2-h training programme aimed at increasing praise for appropriate behaviour. The programme included a feedback session about the pre-test data and a PowerPoint presentation. Data on pupils' time on-task were collected. Furthermore, teachers' job-satisfaction, self-efficacy, and their perceived use of a series of classroom management praxis were investigated through a questionnaire. Social validity data show the teachers within the experimental group to have been highly satisfied with the programme. From observations carried out before and after training, it was seen that changes in teachers' use of feedback were accompanied by increased pupil on-task behaviour. The training did not produce any change in teachers' job-satisfaction scores. Teachers' self-efficacy scores and the frequency of praxis were significantly increased following the training. Implications for practice, limitations, and recommendations for future research are discussed.

**Keywords:** teacher training; praise; classroom behaviour; misbehaviour; teacher self-efficacy

## 1. Introduction

The ability of a teacher to deliver lessons effectively can be detrimentally affected by student misbehaviour. Teachers who continually experience behaviour problems in their classrooms may believe they are ineffective at working with children. Indeed, student misbehaviour is often identified as the main cause of teacher stress [1]. Research has also shown that students' disruptive behaviour may impact teachers to the point of burnout [2]. Burnout has been seen to have a prominent role in predicting future levels of teacher self-efficacy [3], and a teacher's belief of how well they are capable of conducting profession-related activities, such as managing the classroom and using instructional strategies [2]. Elevated teacher stress not only affects the teacher's self-efficacy, well-being and likelihood of burnout, but also the classroom climate [4], which is considered to be a leading indicator of school improvement and a predictor of critical school outcomes [5].

Furthermore, as Witzel and Mercer [6] show, classroom disruptions use up valuable learning time. This suggests troublesome student behaviour impacts learning opportunities and the potential achievement levels of other students. For example, talking out of turn interferes with attending to assigned work, or the teacher, and generally being on-task. This behaviour can therefore be identified as a threat to student learning, as being on-task is necessary for effective learning to occur [7].

It is proposed that increased education and awareness about classroom behaviour, via the application of research-based classroom and behaviour management strategies, may

enable teachers to accurately and appropriately identify troublesome student behaviours and to manage these behaviours more effectively [1]. Strategies aimed at increasing teacher rates of approval have been shown to be effective in decreasing students' inappropriate behaviours and increasing their time on-task, e.g., [8]. Indeed, it was argued by Becker, Madsen, Arnold and Thomas [9] that unless teachers can manage classroom behaviour effectively, their technical teaching skills are wasted. In order to ensure that teachers are sufficiently skilled, so that their time in class is used most effectively and efficiently, the acquisition and ongoing development of classroom management techniques must be promoted.

Behavioural interventions in school classrooms began to have an impact in the 1960s. In the United States, Madsen, Becker and Thomas [10] proved the effect of a package of treatment combining the use of classroom rules, praise and extinction. Several studies using similar methods followed. Most of the investigations detailing behavioural interventions described work carried out with a small number of teachers to improve some aspect of their pupils' performance for a detailed review, see [11]. The main focus of such investigations was on teachers' rate of approval and disapproval. Indeed, in the early days of behavioural interventions, there appears to have been a considerable imbalance between teachers' use of positive and negative feedback [12,13]. From the 1980s onwards, instead, researchers found a reversal of that situation, with more approval than disapproval being given [14–17]. When they analysed their data more closely, however, they found that very much more approval was given to academic behaviours than to social behaviours, a feature White [12] had also noted earlier. More recently, Floress, Jenkins, Reinke, & McKown [18], in one of very few investigations that involved also kindergarten teachers, noticed a change in the general trends of the last three decades. Teacher approval rates across 28 general education classrooms (kindergarten to fifth grade) were collected through direct behavioural observation. Across all grade levels, results suggested that teachers' use of approval was low and teachers used general praise more frequently than behaviour-specific praise.

In the first Italian investigation on the natural rates of teachers' approval and disapproval in the classroom, Sulla, Armenia, and Rollo [19] found that different types of feedback used by teachers were more similar to those found in the earlier investigations carried out in the 1970s than to the types of feedback found in studies from the 1980s to date. In primary schools, the majority of feedback was of a negative nature and directed towards pupils' behaviour (praise:reprimand ratio was 1:3). Most positive feedback was directed in response to pupils' work and very little to pupils' behaviour. In secondary schools, the majority of feedback was of a negative nature and directed in response to pupils' behaviour (praise:reprimand ratio was 1:4). Again, feedback directed towards pupils' social behaviour was very seldom observed.

As well as papers reporting surveys of current practice, there have been a few articles on the training of teachers in the use of behavioural interventions.

Wheldall, Merret and Borg [20] evaluated the Behavioural Approach to Teaching Package (BATPACK), described as "a skill-based package for training teachers" (p. 68). Six primary school teachers were trained in the use of BATPACK during six weekly one-hour sessions, with reading assignments between sessions. In their classes, both on-task behaviour and the teachers' positive responses to their students increased. Negative responses remained more or less unchanged in the experimental group, while for the control group, negative responses more than doubled during the same time period.

Another training programme which received considerable publicity was the American programme Assertive discipline by Canter [21]. Like BATPACK, it is based on behavioural principles. It requires six hours of training and includes many examples of good practice shown to teachers on video. The effectiveness of the programme has been demonstrated by a number of evaluative studies, for instance those of Nichols and Houghton [22], Swinson and Melling [23] and Woods, Hodges and Aljunied [24].

In each of the programmes described, there was a significant investment of time. In England, Swinson and Harrop [25] made an attempt to produce similarly successful

outcomes while decreasing the investment in training time. A group of teachers took part in a brief, one session in-service course, in which they were trained in behavioural techniques with the main aim of helping them increase their rates of positive statements contingent upon pupils' required behaviours, and to decrease their rates of negative responses. The training took place in the evening and lasted approximately two-and-a-half hours. The training basically consisted of feedback to the teachers, based on a preliminary analysis of the pre-training lessons, and a PowerPoint presentation called 'Managing behaviour—four essential steps'. As a result, the teachers showed increased levels of positive statements contingent upon appropriate classroom behaviour and decreased levels of negative statements; these changes being accompanied by increased student on-task behaviour.

Sulla and Perini [26] evaluated the effects of the cueing system and of visual performance feedback on two Italian primary school teachers' use of approval. The intervention resulted in a decrease in teacher disapproval. Additionally, students' time on-task increased.

Yet, when it seems to be a critical point, apart from Sulla and Perini [26], there is a general lack of research in Italy about teacher training in the use of approval in classroom. This study aimed to add to the literature on this subject.

*Current Study*

The study aimed to:
- Investigate the effect of training teachers to alter their verbal feedback and become more positive;
- Investigate the effect that changes in teachers' verbal behaviour have on the on-task behaviour of their students;
- See whether any changes in teachers' use of feedback would have an effect on their job satisfaction, self-efficacy, and their perceived use of a series of behavioural classroom management praxis.

## 2. Materials and Methods

### 2.1. Setting and Participants

In this investigation, teaching staff from a primary school in a region in Northern Italy took part in a training study. The sample consisted of 32 primary school teachers (2 males and 30 females) aged between 33 and 62 years of age (M = 46, SD = 7.46). The teachers in our sample were all tenured teachers with the number of years teaching ranging from four to 40 (M = 16.63, SD = 8.81). 41.7% had a high school diploma, 54.2% had a degree, and 4.2% did not report their title. The whole staff were part of the original phase of the aforementioned investigation aimed at examining the natural rates of teacher approval and disapproval in Italian school classrooms [19]. After that, they took part in a 30 min meeting, during which they received feedback on the initial set of observations. Sixteen out of thirty-two teachers volunteered to take part in the training. All members of staff took part in the second set of observations. Therefore, the teachers who took no part in the training were considered the control group. The differences in the main characteristics taken into consideration (i.e., gender, age, and years of teaching) between the experimental and the control group were not statistically different. There were one male and 15 female teachers in each group. The difference in age between the experimental group (M = 47.6; SD = 6.38) and the control group (M = 44.5; SD = 8.29) was not statistically different ($t_{(30; 28 \cdot 156)} = -1.218$; $p = 0.233$). Although the teachers in the experimental group were more experienced than the teachers in the control group, the difference in the years of teaching between the experimental group (M = 18; SD = 10.18) and the control group (M = 14; SD = 6.35) was not statistically different ($t_{(30; 25 \cdot 143)} = -1.833$; $p = 0.079$).

## 2.2. Instruments

- MICRO (Mixed Interval Class Room Observation). The MICRO [27] "entails a repeated timed observation being made of a small sample of five randomly chosen students in a class. Observations are alternated with recordings of teacher behaviour. The students are observed as being on-task or off-task" (p. 14). Teachers' verbal behaviours are tallied under four headings: Task Performance Positive comments (TPP); Social Behaviour Positive comments (SBP); Task Performance Criticism (TPC); and Social Behaviour Criticism (SBC).

- MESI Questionnaire. Three out of the six questionnaires contained in the MESI [Motivazione, Emozioni, Strategie e Insegnamento: Motivations, Emotions, Strategies and Teaching] battery of tests [28] were used to investigate teachers' job satisfaction, self-efficacy, and the use of a series of praxis about classroom management, before and after the training.

  Job satisfaction and self-efficacy

  A considerable amount of literature attests to the positive influence that both self-efficacy beliefs and job satisfaction have on performance in a variety of settings, including schools [29,30]. It seems that teachers with high self-efficacy are more likely to manage classroom problems successfully [31], and to keep students on-task [32]. Job satisfaction was evaluated via a five-item questionnaire which employed a Likert rating scale with end points of one and seven (1 = Strongly disagree; 7 = Strongly agree). Self-efficacy was evaluated via a 24-item questionnaire, which was the Italian translation of a questionnaire by [33]. The questionnaire measured teacher's self-efficacy perception in several situations linked with teaching and classroom management. Responses were made on a Likert-type rating scale with end points of one and nine (1 = Not at all; 9 = Very much).

  Praxis

  The questionnaire measured the frequency with which teachers apply a series of praxis when teaching. The questionnaire consists of 25 items. Responses were made on a Likert rating scale with end points of one and five (1 = Never; 5 = Always).

- Social Validity Questionnaire. Social validity refers to the degree that behaviour-change efforts impact favourably upon consumers [34]. To assess social validity, a 10-item questionnaire was created by the authors, which employed a Likert rating scale with end points of one and five (1 = Strongly disagree; 5 = Strongly agree). The questionnaire was administered in paper/pencil format. The questionnaire investigated teachers' perceptions of the training's social relevance, and the acceptability of its goals, procedures, and outcomes.

## 2.3. Dependent Measures

As in the most recent investigations [17,35,36], exclusively verbal behaviour was considered. Non-verbal behaviour is harder both to define and observe. That makes it difficult to obtain reasonably high levels of inter-observer agreement [37]. Definitions of target behaviours as reported in the overview and guidance notes of the MICRO [27] were (p. 15).

TPP (Task Performance Praise): Enthusiastic or positive recognition/praising comments addressed to students about outcomes from a specified activity that has been directed, organised or sanctioned by the class teacher; SBP (Social Behaviour Praise): Enthusiastic or positive recognition/praising comments to students in respect of their pro-social behaviours or compliance with instructions or rules that an adult has given them; TPC (Task Performance Criticism): Critical or corrective comments to students about outcomes from a specified academic activity that has been directed, organised or sanctioned by the class teacher; SBC (Social Behaviour Criticism): Corrective comments and repeated directions addressed to students about anti-social, non-compliant or unacceptable behaviours by an adult.

Data on children's on-task behaviour were also collected. Pupils were considered as being on-task when actively engaged (e.g., looking at or writing on) in independent

seatwork, teacher instruction, designated classroom activities, and/or engaging in task-related (permissible) vocalisations with teachers and/or peers. Pupils' heads had to be oriented toward the teacher or task. If items were involved in the relevant task demand, the pupils had to actively be using that item.

*2.4. Procedure*

A letter presenting the project was sent to the head teacher. The head teacher allowed observations to be made in the school. The initial set of observations took place in the school three weeks prior to the training and represented the pre-test measure. Observations were conducted via direct observations by the first author and a trained doctoral-level psychology graduate student, and occurred between four and six times per week. In the current study, the second observer was trained in the use of the proforma using brief tapes showing teacher-pupils interactions in classroom. Findings of prior investigations or expectations from the present study were not mentioned. This procedure was continued until the percentage of observer agreement reached above 80% on two successive occasions. From that point, the two observers entered the classrooms and scored the actual lessons. The observation procedure did not start immediately on their arrival in the classroom. Students were allowed to come into the room and find their seat. Once the class had begun to settle down, the scoring of the teacher's statements and the behaviour of the students started. When observing the class, each observer independently sat at the back of the room in a spot where they could observe all the students—usually the classroom's back corners.

Pupils' on-task rate was recorded using a momentary time sampling method. That is, at the end of a 1-min interval, observers looked at a student in a predetermined order and indicated whether they were following the teacher's directions. The observer then observed the next student in the same manner. Once all the pre-selected five students had been observed, the observers could focus on recording the verbal behaviour of the teacher for the rest of the 1-min interval.

The teachers' rate of TPP, SBP, TPC, and SBC were recorded using an event recording procedure in which their frequencies within 1-min interval were recorded, converted to a rate-based measure, and reported as count per minute during 30-min observation sessions.

These observation procedures were utilised across all phases.

One week after the end of the initial set of observations, and ten days before the next phase, the entirety of the teaching staff (32 teachers) participated in a 30-min meeting held by the authors and received feedback about the observations. During this ten-day period of time, teachers voluntarily enrolled into the training by sending an e-mail to the first author. After that, 16 teachers took part in a 90-min meeting.

The second set of observations of the whole teaching staff took place between three and five weeks after the training, following the procedure outlined in the pre-test measure, and represented the post-test measure.

Questionnaires assessing job satisfaction, self-efficacy, and use of praxis, were administered to the teachers both during the first observation (pre-test) and one week after the post-test—between four and six weeks after the training.

Training

The training of the teachers was conducted by the first author. The experimental group attended a (1) feedback session (of 30-min duration) and (2) viewed a PowerPoint presentation (of 60-min duration). The control group only participated in the feedback session (i.e., 1).

1. Feedback session. For the first element, results of the questionnaires on job satisfaction, self-efficacy, and the use of praxis were presented to the teachers by the authors. Mean scores only for the whole staff were presented to the teachers, not individual scores. Although the teachers were made aware from the project presentation letter that they had a right to see the record of observations at the end of each observation day, nobody requested to see it. The feedback on the teachers' current use of verbal feedback was based on a

preliminary analysis of the original sets of pre-training scorings. The identity of individual teachers was kept confidential. Findings were reported only on the basis of the whole school's results and were reported back only in terms of percentages of feedback given. Comparisons were made with previous research in this area, particularly the work of Apter et al. [17] on British primary schools. A histogram was shown to the teachers that compared both the data collected in their school and the data collected in the rest of Italy (2019), as well as the data found by Apter et al. [17] about primary school teachers' verbal behaviour and their students' on-task time in the United Kingdom. The histogram showed quite clearly the difference between the Italian sample and British sample: in the latter, there was a higher rate of approval, and a much lower rate of disapproval. Furthermore, the British primary school students' time on-task was significantly higher. However, it was pointed out to the teachers that their current teaching style was essentially a reactive one seeing that much of their feedback, especially their negative responses to social behaviour, was in response to students that basically were not doing as they were asked to. It was explained that reprimanding students was essentially a very limited strategy, which only produced short-lived changes in their behaviour. It was argued that a much more proactive strategy, one that involved providing a great deal more in terms of positive feedback, especially positive feedback aimed towards social behaviour might prove to be a much more effective way of leading to improved pupil behaviour. Functional analysis of disapproval was presented via several practical examples (videotapes) in order to emphasise that pupils' disruptive behaviour is usually fuelled by the teacher's negative attention. When the teachers discussed their schools' results and the remarks made about teaching styles, there was general agreement that the views expressed about being more proactive seemed logical. There were no major objections raised.

2. PowerPoint. The second element and core of the training consisted of the revised version of the PowerPoint presentation 'Managing behaviour—four essential steps' [25]. Compared to the original one, the revised version included an introductory part on the functional analysis of teacher approval and disapproval. The presentation included 50 slides. Most of the slides included graphics aimed at representing the point being made and included very succinct written material. The teachers were provided with a copy of the presentation and were encouraged to make notes. After that, the four essential aspects of the original PowerPoint were presented. These may be summarized in the following way [25] (p.120): 1. Always make your requirements absolutely clear; 2. Remember to look for the behaviour you want rather than the behaviour you do not want; 3. Frequently acknowledge pupils when they are doing what is required; and 4. Change the frequency of the feedback to suit the situation.

### 2.5. Inter-Observer Agreement

The first author and a trained doctoral-level school psychology graduate student collected inter-observer agreement data on 96% of sessions. Agreement rates were calculated using the Kappa coefficient [38]. Kappa was calculated at between 0.77 and 0.89 for joint observations, with a mean value of 0.85.

### 2.6. Data Analysis

Assumptions of normality were met, therefore a repeated-measures mixed analysis of variance (group X time: control vs experimental X before vs after the training) was utilised to detect any change in the rate per minute of teachers' verbal behaviours; average pupils' time on-task; and teachers' scores in job-satisfaction, self-efficacy, and praxis. Descriptive statistics for the Social Validity Questionnaire results are provided.

## 3. Results

### 3.1. Teacher Feedback

The changes in the teachers' verbal behaviour as a result of their training are reported in Table 1 and Figures 1–4.

**Table 1.** Average rate per minute of different type of teacher verbal feedback before and after the training: Means (SDs).

| Group | Praise | | | | Criticism | | | |
|---|---|---|---|---|---|---|---|---|
| | Task | | Social | | Task | | Social | |
| | Pre | Post | Pre | Post | Pre | Post | Pre | Post |
| Experimental | 0.22 (0.1) | 0.49 (0.3) | 0.01 (0.01) | 0.11 (0.1) | 0.15 (0.1) | 0.06 (0.1) | 0.64 (0.2) | 0.39 (0.2) |
| Control | 0.21 (0.1) | 0.15 (0.1) | 0.01 (0.02) | 0.01 (0.03) | 0.15 (0.1) | 0.19 (0.2) | 0.61 (0.3) | 0.64 (0.3) |

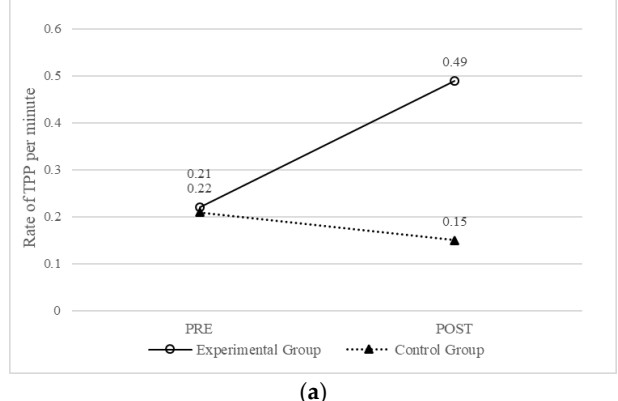 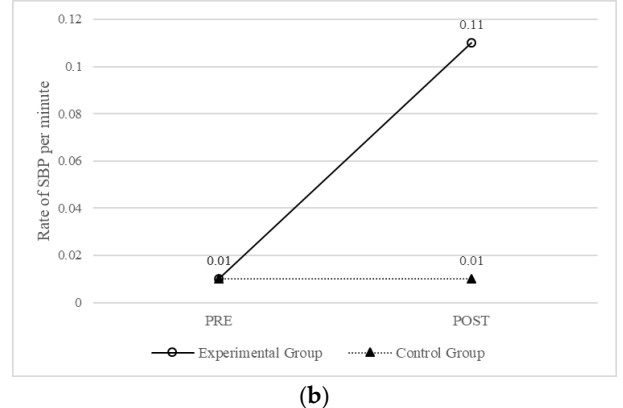

(**a**) (**b**)

**Figure 1.** (**a**) Teacher Task Performance Praise (TPP) rate/min before and after the training; (**b**) Teacher Social Behaviour Praise (SBP) rate/min before and after the training.

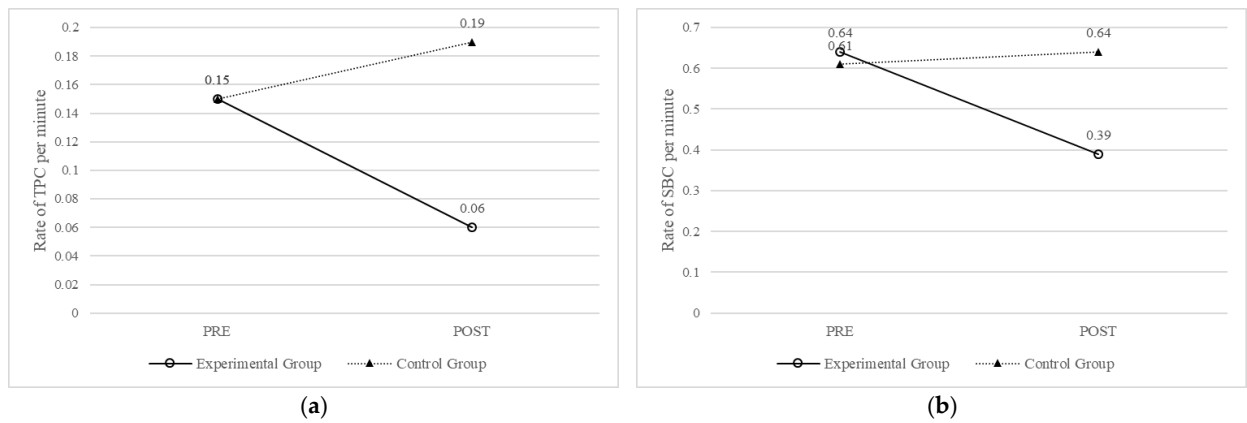

(**a**) (**b**)

**Figure 2.** (**a**) Teacher Task Performance Criticism (TPC) rate/min before and after the training; (**b**) Teacher Social Behaviour Criticism (SBC) rate/min before and after the training.

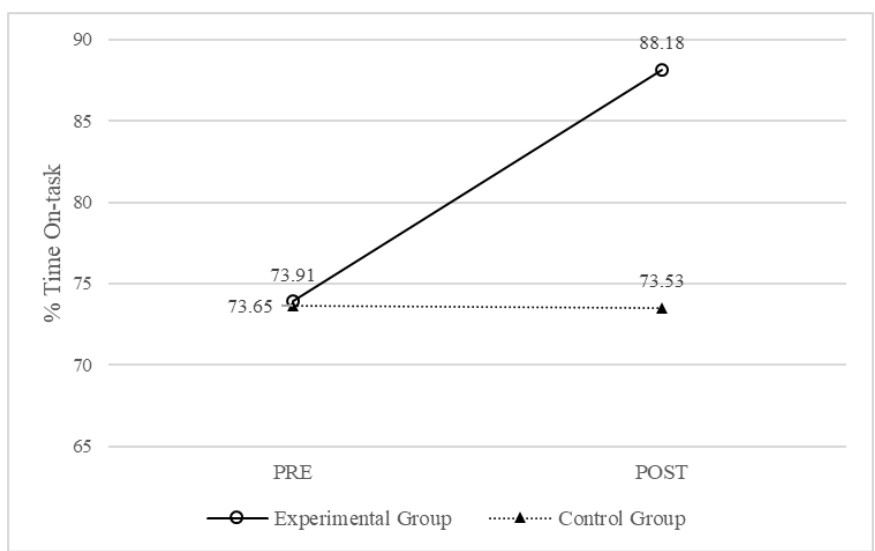

**Figure 3.** Students' time on-task by group.

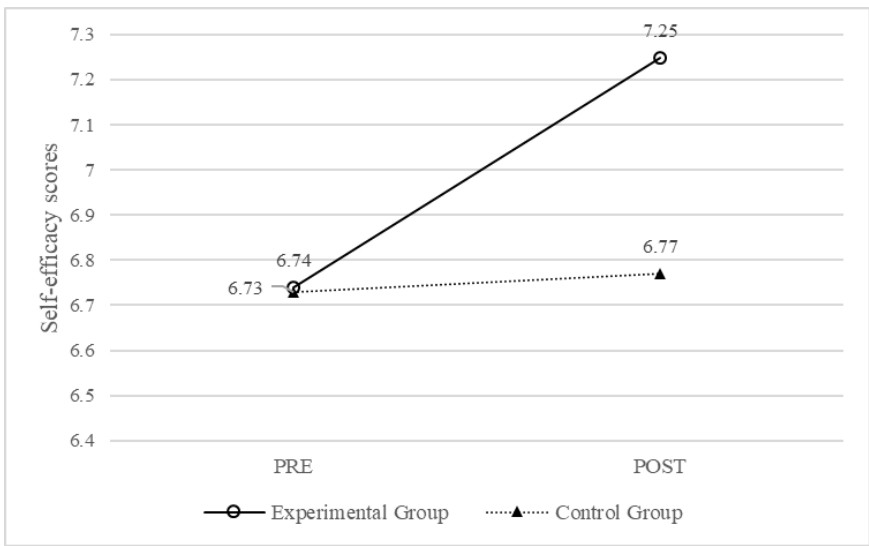

**Figure 4.** Teachers' self-efficacy scores before and after the training.

Mixed repeated-measures analysis of variance (group per time) revealed that the training produced a post-test significant increase both in task performance praise (significant interaction, Figure 1a: $F_{(1,30)} = 10.963$, $p < 0.01$, $\eta^2 = 0.27$) and in social behaviour praise rate per minute (Figure 1b: $F_{(1,30)} = 10.121$, $p < 0.01$, $\eta^2 = 0.25$) only for the teachers in the experimental group.

Symmetrically, the same ANOVA testing the interaction group per time on teacher criticism showed a statistically significant decrease at post-test only for the teachers in the experimental group, both for criticism to task performance (Figure 2a: $F_{(1,30)} = 11.228$, $p < 0.01$, $\eta^2 = 0.27$) and criticism for social behaviour (Figure 2b: $F_{(1,30)} = 13.034$, $p < 0.01$, $\eta^2 = 0.30$).

*3.2. Student Behaviour*

The changes in pupil behaviour went in the expected direction as a result of the training (Figure 3): a mixed repeated-measures analysis of variance revealed that the training produced a significant increase in students' time on-task at the post-test ($F_{(1,30)} = 25.793$, $p < 0.001$, $\eta^2 = 0.46$).

*3.3. MESI Questionnaire*

Job satisfaction. The means and standard deviations for job satisfaction scores on pre- and post-test are shown in Table 2. A mixed repeated-measures analysis of variance revealed that the training did not produce a significant change at post-test.

**Table 2.** Scores in teacher job satisfaction before and after the training: Means (SDs).

| Group | Pre-Test | Post-Test |
|---|---|---|
| Experimental | 5.53 (0.8) | 5.57 (0.8) |
| Control | 5.51 (0.4) | 5.51 (0.4) |
| Total | 5.52 (0.6) | 5.54 (0.6) |

3.3.1. Self-Efficacy

A repeated-measure analysis of variance (Figure 4) revealed that the training produced a significant increase in self-efficacy scores at the post-test ($F_{(1,24)} = 7.401$, $p < 0.05$, $\eta^2 = 0.24$).

3.3.2. Praxis

A repeated-measure analysis of variance showed (Figure 5) that only the experimental group's scores increased significantly at the post test ($F_{(1,24)} = 5.896$, $p < 0.05$, $\eta^2 = 0.20$).

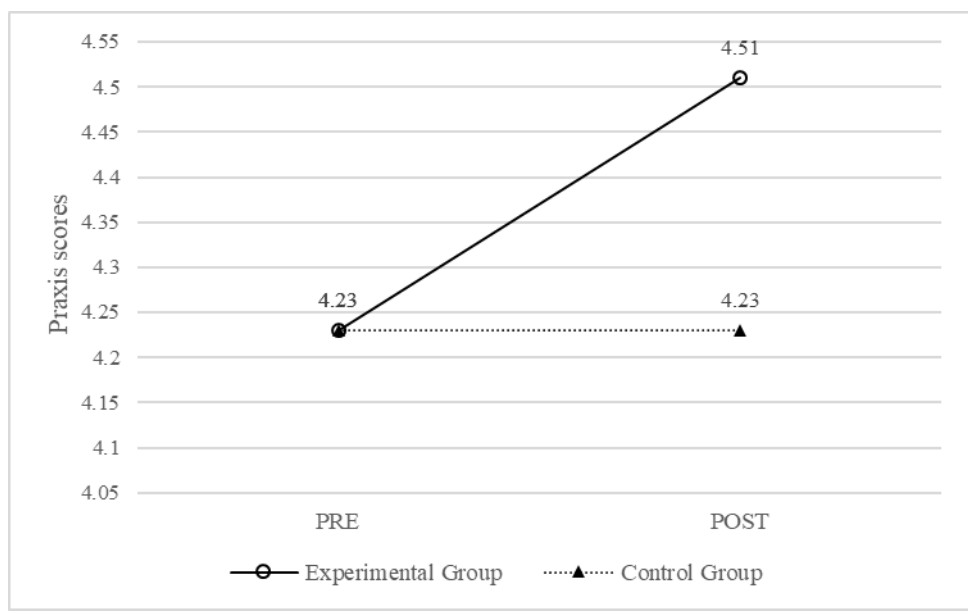

**Figure 5.** Teachers' praxis scores before and after the training.

A negative relationship was found between the score in praxis and teacher average disapproval rate per minute (r = −0.420, $p < 0.05$). The result was expected given that, among others, the questionnaire about teacher praxis includes items such as "I positively reinforce student's effort and good results".

*3.4. Social Validity*

Results of the social validity questionnaire that was given to participants are displayed in Figure 6. Questionnaires were given to teachers at the end of the last observation and re-collected a week after. Due to contingent issues, the questionnaires on social validity completed by four teachers in the experimental group could not be collected. The results of 11 out of 15 teachers who took part in the training are displayed in this paper.

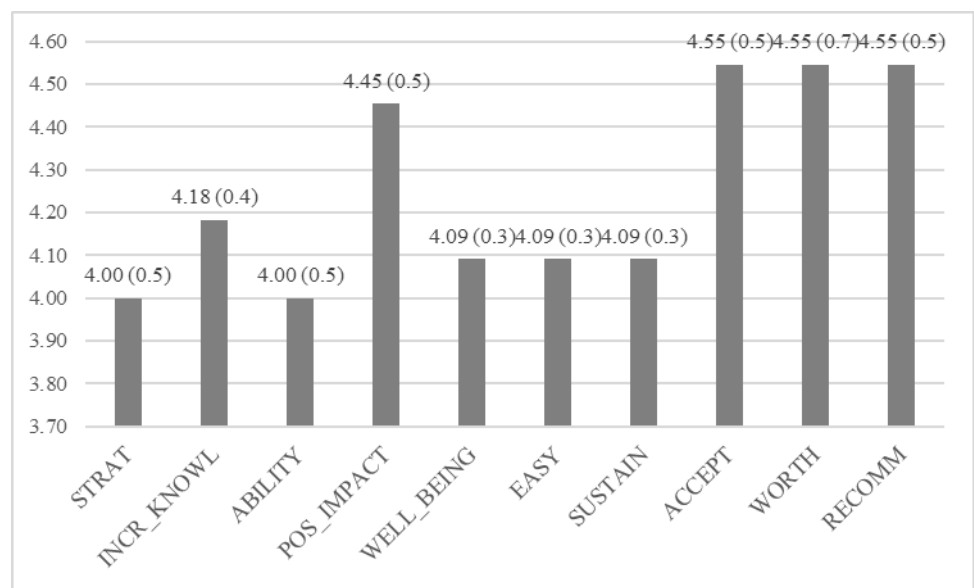

**Figure 6.** Results of the social validity questionnaire: Means (SDs).

The data show the teachers within the experimental group to have been highly satisfied with the programme. The mode response on the majority of the questions was 5 (strongly agree). Ninety per cent agreed or strongly agreed that the programme made a positive impact on their well-being (M = 4, SD = 0.5), and 100% agreed or strongly agreed that the programme improved positive school outcomes (M = 4.45, SD = 0.5). Likewise, on Question 2 (M = 4.18, SD = 0.5) and Question 3 (M = 4.09, SD = 0.30), 100% of teachers agreed or strongly agreed that they had increased their knowledge and skills in classroom management.

Concerning the procedures for the program's interventions, 100 per cent of teachers agreed or strongly agreed that procedures were easy (M = 4.09, SD = 0.3). In addition, 100 per cent of teachers agreed or strongly agreed that the strategies we presented were sustainable (M = 4.09, SD = 0.3). The majority of the teachers (M = 4.55, SD = 0.5) also agreed or strongly agreed that they would recommend the programme to other educators, and 100% (M = 4.55, SD = 0.7) agreed or strongly agreed that the programme was worth their time and effort.

## 4. Discussion

The main goals of the training programme presented in this paper were: to help teachers improve their rates of approval; to reduce their rates of disapproval; and to increase their students' on-task behaviour. The results show that the programme has effectively achieved those aims. Both teachers' approval and disapproval rates were changed by a relatively brief intervention and these changes were accompanied by increased levels of pupils' on-task behaviours. Because the investigation comprised one pre-training measure and one post-training measure, long-term effects of the intervention cannot be claimed. One may assume that the increased level of student on-task behaviour that was experienced by all teachers in the sample must have laid stress on the value of maintaining their approval/disapproval rates at their post-training levels. However, this cannot be taken for granted.

As in the rest of the Italian schools [19], in this investigation, the majority of teachers' feedback was of a negative nature and directed in response to students' social behaviour. Positive feedback was mainly directed toward students' academic behaviour, while feedback directed towards their social behaviour was very seldom observed. The intervention successfully changed this balance. Before training, teachers were using three times as much disapproval than approval, while after the programme this relationship was reversed and

more approval than disapproval was given. The rate of approval per hour was almost tripled. Approval for social behaviour—that was nearly non-existent—became ten times more frequent than it was before the intervention.

It is worth noting that such results were obtained after training that took only two hours to deliver. It is therefore worth considering the elements of training used in the intervention that made it so powerful. First, the advice given to the teachers was evidence-based. This allowed the presenter to state that 'we recommend you treat students in this way because we have sound evidence that if you do, it will work', rather than presenting a series of bland suggestions the teachers might like to try [25]. Second, at the beginning of the presentation teachers received feedback on their current use of verbal behaviour, as recorded in the pre-training observation. The results for their school were similar to those found in the Italian sample, allowing the presenter to highlight the underuse of positive feedback and the overuse of negative feedback. Above all, the results for the British sample had the opposite tendency, which reflected a better student conduct. This represented further evidence in favour of the strategies that were suggested to them. Third, every attempt was made to keep the content of the presentation as simple as possible. Changing teachers' values is the most difficult aspect because they have become well-ingrained. It can be perceived that presenting easy-to-use techniques may represent a winning strategy. Furthermore, seeing as we were only able to give the training to teachers at the end of an eight-hour school day, a longer and more complex intervention would probably have not been as well received as the one we proposed. These are the main reasons why an intervention very similar to the one used by Swinson and Harrop [25] was chosen. Indeed, teachers appeared to understand the simple message contained in the "*4 essential steps*" and, as the results at the post-training observations showed, they were implementing the strategies that had been offered to them in their classrooms. The teachers' newfound skills produced an increase both in the use of praxis and in teachers' self-efficacy. Higher scores in "praxis" may well be a consequence of increased knowledge and skills due to the training. The experience of student behaviour change due to changes in the teachers' strategies may have had a positive impact on teachers' self-efficacy. The fact that teachers' job satisfaction was not affected by changes due to the training they received may be a consequence of the fact that other variables that influence job satisfaction were not take into account (e.g., working conditions, relation with colleagues, etc.).

*Limitations and Directions for Future Research*

This study incudes limitations and directions for future research that warrant discussion. First, this study only included the teaching staff of a single primary school. As a result, the generalisability to other teachers may be limited. Hence, future research should seek to replicate these procedures within larger groups of educators and across other grades, possibly utilising randomised controlled trials in order to reduce possible bias and ensure cause-effect relationships between teacher training and students' outcomes. Some successful experiences are reported in the literature. However, to the best of our knowledge, only in kindergarten [39] and primary schools [40]. Second, due to the end of the school year, and the concomitant end of the doctoral programme of the first author, follow-up data could not be collected. In the near future, a follow-up investigation should evaluate whether the changed levels in both teacher and pupil behaviour were maintained. Nevertheless, a generalisation programme must be planned in advance and carried out in parallel to the training programme, seeing that we cannot take maintenance of the acquired skills for granted. Indeed, in a recent study, Simonsen et al. [41] explored the effect of a brief training programme on teachers' use of several classroom management skills including specific praise. Teachers in their study increased their specific praise rates while they were actively engaged in the course. However, the change did not sustain during a maintenance phase when participants shifted their focus to another skill. Moreover, it would be valuable to verify any improvements in teacher job satisfaction following training programs on classroom management. Third, it is worth noting that social behaviour criticism was significantly

decreased, although it persisted post training. In the future, multi-component consultation strategies, including a direct specific intervention for the reduction of disapproval, as well as measures of treatment integrity [42], could promote further reduction—and even the neutralisation—of social behaviour criticism in the classroom. Fourth, the current study only considered the teachers' perspective. It is well established in the literature, e.g., [43] that most students prefer to receive approval, and that students who perceive that they frequently receive ability and effort feedback from their teacher report more satisfaction with the classroom environment, and a positive teacher-student relationship. As we know, a supportive school climate is critical for school effectiveness [43]. Future investigations should assess students' personal perceptions of the classroom climate before and after training aimed at increasing teachers' praise, in order to further validate the results of the current investigation.

**5. Conclusions**

In conclusion, the findings from this study show that the short course of training achieved its primary aims of increasing teachers' rates of approval and of decreasing disapproval, with concomitant improvements in their students' behaviour, and improving teachers' self-efficacy. However, future research is needed in order to address this study's limitations.

**Author Contributions:** Conceptualization, F.S. and D.R.; methodology, F.S. and D.R.; data curation, F.S.; writing—original draft preparation, F.S.; writing—review and editing, F.S.; supervision, D.R.; project administration, F.S. All authors have read and agreed to the published version of the manuscript.

**Funding:** This research received no external funding.

**Institutional Review Board Statement:** The study was conducted in accordance with the Declaration of Helsinki, and approved by the Institutional Review Board of Parma University (protocol code 72, 21/04/2014).

**Informed Consent Statement:** Informed consent was obtained from all subjects involved in the study.

**Data Availability Statement:** The data presented in this study are available on request from the corresponding author. The data are not publicly available due to privacy reasons.

**Acknowledgments:** The authors want to thank Eusebia Armenia for the help in data collection.

**Conflicts of Interest:** The authors declare no conflict of interest.

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
