# Peer review of "The Effect of a Short Course on a Group of Italian Primary School Teachers’ Rates of Praise and Their Pupils’ On-Task Behaviour"

_education, doi:10.3390/educsci13010078_

Round 1

Reviewer 1 Report

This article describes the effects of a brief 2 hour long instructional program for Italian primary school teachers. The program taught the teachers how and why to praise appropriate behavior of students. This led to changes in the teachers' behavior towards their students in that regard in accordance with the instructions given. Moreover, those increased rates of praise by teachers correlated with increases of on-task behavior in their students. This was clear when their data was compared to pretests and to a comparison group of teachers who did not participate in the program and the on task behavior of their students. The manuscript is very well written, it is clear, concise, and describes a successful program. Social validity measures showed the teachers in the experimental group to be satisfied and measures of self-efficacy increased. Dependent measures were collected with an appropriate and formerly tested system for direct observation of pupils and teachers behavior in the classroom. Inter-observer agreement ratings were sufficiently high. The collected data was statistically analysed for the whole group in each case and measures of praise and on-task behavior were presented as rate per minute for all phases of the study. Evaluation of changes in the behavior of teachers and students in both experimental and comparison groups based on the direct observations are clearly shown and are easily understandable. Other measures taken are also clearly displayed. The impact of the program is clear. The authors acknowledge the fact that replications are needed to learn about the replicability and reliability of the results and thus their generalisability to other schools and teacher groups. The authors also acknowledge that their study cannot answer the the question about the long term effects of their brief instructional program and replications of the study should look at the long-term impact. I totally agree. I also am of the opinion that direct observation of behavior to measure procedural integrity is preferable to self-report measures. I find the article merits publication as is and I find its content very practical and important for educational research and for teachers in general and for those who work with them. I only have two minor comments for change. On line 200, insert "be" before "using" and in lines 485 and 486 the style of the reference is not in sync with the style of all the other references.

Author Response

Dear Reviewer1,

on behalf of my co-author, we wanted to thank you very much for taking the time to read our manuscript in depth and giving such positive and uplifting feedback.

We made the requested revisions – that are marked up in the new version of the manuscript using the Track Changes function (please, see line 200, lines 485 and 486 - now 488 and 489).

Once again, thank you very much for the comments and suggestions.

Reviewer 2 Report

The study investigated the effect of teacher training on job satisfaction and self-efficacy. The pre-and post-test design is appropriate and the tables and figures are helpful for interpreting the main findings. The limitation section addressed some legitimate concerns regarding the scope and implications of the study and the future directions of this research. The literature review section could benefit from citing more recent empirical studies using experimental design (especially RCT) to investigate the impact of teacher training on teacher retention, job satisfaction, and the indirect effects on students' learning experiences as relating to the future directions of this work when the impacts on both teacher and student are considered within the same model. 

Author Response

Dear Reviewer2,

on behalf of my co-author, we wanted to thank you for the valuable comments on our manuscript. We do agree with your suggestion and we have made revisions accordingly – that is marked up in the new version of the manuscript using the Track Changes function (please, see lines: 429-434; 555-560).

Once again, thank you very much for the positive comments and suggestions.